# Cumulative Effects of Particulate Matter Pollution and Meteorological Variables on the Risk of Influenza-Like Illness

**DOI:** 10.3390/v13040556

**Published:** 2021-03-26

**Authors:** Kacper Toczylowski, Magdalena Wietlicka-Piszcz, Magdalena Grabowska, Artur Sulik

**Affiliations:** 1Department of Pediatric Infectious Diseases, Medical University of Bialystok, Waszyngtona 17, 15-274 Bialystok, Poland; artur.sulik@umb.edu.pl; 2Department of Theoretical Foundations of Biomedical Sciences and Medical Computer Science, L. Rydygier Collegium Medicum in Bydgoszcz, Nicolaus Copernicus University in Torun, 9 M. Sklodowska-Curie St., 85-094 Bydgoszcz, Poland; mpiszcz@cm.umk.pl; 3Parexel International, Zwirki i Wigury 18A, 02-092 Warsaw, Poland; 4Department of Hydrobiology, University of Bialystok, Ciolkowskiego 1J, 15-245 Bialystok, Poland; magra@uwb.edu.pl

**Keywords:** particulate matter, air pollution, influenza, influenza-like illness, COVID-19

## Abstract

The cold season is usually accompanied by an increased incidence of respiratory infections and increased air pollution from combustion sources. As we are facing growing numbers of COVID-19 cases caused by the novel SARS-CoV-2 coronavirus, an understanding of the impact of air pollutants and meteorological variables on the incidence of respiratory infections is crucial. The incidence of influenza-like illness (ILI) can be used as a close proxy for the circulation of influenza viruses. Recently, SARS-CoV-2 has also been detected in patients with ILI. Using distributed lag nonlinear models, we analyzed the association between ILI, meteorological variables and particulate matter concentration in Bialystok, Poland, from 2013–2019. We found an exponential relationship between cumulative PM_2.5_ pollution and the incidence of ILI, which remained significant after adjusting for air temperatures and a long-term trend. Pollution had the greatest effect during the same week, but the risk of ILI was increased for the four following weeks. The risk of ILI was also increased by low air temperatures, low absolute humidity, and high wind speed. Altogether, our results show that all measures implemented to decrease PM_2.5_ concentrations would be beneficial to reduce the transmission of SARS-CoV-2 and other respiratory infections.

## 1. Introduction

Particulate matter (PM) is an important air pollutant. It comprises multiple components and size fractions. The size of the inhalable particles is limited to those with an aerodynamic diameter of 10 μm or less (PM_10_). PM_10_ includes fine particles with aerodynamic diameters of 2.5 μm or less (PM_2.5_) and coarse particles with aerodynamic diameters between 2.5 μm and 10 μm (PM_10-2.5_). PM_2.5_ originates primarily from combustion sources, while PM_10-2.5_ is produced by agriculture, mining, construction activities and road dust resuspension [1]. The health effects of particulate matter may differ because of differences in chemical composition and penetration into the respiratory tract [2]. Smaller particles have been shown to be more harmful to human health than larger particles because of the potential to be deposited in the alveoli and the ability to cross the lung–blood barrier [3]. The evidence for airborne particulate matter and its public health impact consistently shows adverse health effects at exposure levels that are currently experienced by urban populations in both developed and developing countries [1]. The range of health effects is broad. Exposure to air pollutants has been shown to be the cause of increased emergency room visits, exacerbation of chronic respiratory and cardiovascular diseases, decreased lung function, and premature mortality [2]. In response to the accumulating evidence on the adverse effects of air pollution, WHO published air quality guidelines with daily and annual thresholds for mean concentrations of PM_2.5_ and PM_10_. The guidelines stipulate that PM_2.5_ concentrations should not exceed a 25 µg/m^3^ daily mean or 10 µg/m^3^ annual mean. For PM_10_, concentration thresholds are 50 µg/m^3^ and 20 µg/m^3^ for daily and annual means, respectively [1].

Based on accumulating evidence, PM increases the incidence of influenza and influenza-like illness [4,5,6]. However, because of the difference in size and the ability to penetrate deep into the lungs, the effect might be stronger for PM_2.5_ than for PM_10_ [7]. Influenza-like illness (ILI) is a common acute respiratory syndrome defined by WHO as fever (>38 °C) with cough or sore throat, which began in the last 10 days [8]. Each year, ILI imposes a significant burden on public health and leads to substantial morbidity and economic costs [9,10].

According to the latest WHO database of observed annual mean PM_2.5_ concentrations, air quality in Poland is considered unsafe, with many of Europe’s 50 most polluted cities located in Poland [11]. High pollution levels in Polish cities are caused mainly by coal power plants and residential heating with wood and coal [12]. Following the recognition that Poland is facing high air pollution levels, researchers have focused on the associated health effects. However, little is known about the influence of air pollution on the respiratory infection incidence in Poland. In a recent study, Slama et al. reported a positive association between air pollutants and hospitalizations for respiratory disease in Polish cities [13]. This, and other studies have shown that people exposed to air pollutants may be at high risk for infection within a lag period of zero to ten days, which represents the usual incubation periods of respiratory infections [4,14,15]. Little is known about the effect of air pollution on respiratory infections beyond a lag of two weeks.

The current COVID-19 outbreak has been caused by the emergence of a new coronavirus, SARS-CoV-2, that very quickly spread in the human population. A similarity between COVID-19 and ILI symptoms has been observed. The presence of SARS-CoV-2-positive swabs in patients with ILI has been reported in several studies [16,17,18]. Moreover, air pollution was suggested to be a major factor contributing to the aggressive spread of this virus [19]. Meteorological variables play an important role in the physical and chemical transformation of PM and its horizontal and vertical transport [20]. Therefore, we aimed to assess the association between the risk of ILI and air pollution over longer time lags, considering the potential influence of meteorological variables and long-term trends.

## 2. Materials and Methods

### 2.1. Influenza-Like Illness Data

Bialystok is the largest city in northeastern Poland and one of the largest cities in Poland in terms of population density (2913.8 people per sq. km). Bialystok City and Bialystok County cover an area of 3086.76 square kilometers and are inhabited by approximately 435,000 people, 297,000 (68%) of whom live in Bialystok City [21]. The city has a warm, humid continental climate [22].

The data on the number of suspected influenza and influenza cases in Bialystok City and Bialystok County (Figure 1) were obtained from the Department of Epidemiology of the Municipal Sanitary and Epidemiological Station. Influenza is a notifiable disease in Poland. The Department collects all reports of confirmed and suspected influenza cases that come from all hospitals and all outpatient clinics located in the city and county. Reports of confirmed and suspected influenza cases (ICD-10 codes: J10.0, J10.1, J10.8, J11.0, J11.1, and J11.8) are submitted manually by hospitals and clinics with no automated reporting system integrated with medical document management software. The data are then reported weekly by the Municipal Sanitary and Epidemiological Station. No data on the percentage of confirmed cases in the total reported number are available. Therefore, throughout the manuscript, both confirmed and suspected influenza cases are analyzed jointly as ILIs.

### 2.2. Meteorological and Air Pollution Data

The data on the concentration of air pollutants were obtained from two fixed stations, Waszyngtona Street and Warszawska Street, which are located two kilometers apart in the city center and have a maximum distance to the city borders of 6 km. The concentrations of PM_2.5_ were measured with BAM 1020 (Met One Instruments Inc., OR, USA) (Waszyngtona Street), and the concentration of PM_10_ was measured with TEOM 1405F (Thermo Fisher Scientific, MA, USA) (Warszawska Street). Daily meteorological data (air temperature, relative humidity, wind speed, precipitation, precipitation and sunshine duration) for the analyzed period were obtained from the Institute of Meteorology and Water Management. Absolute humidity was calculated from temperature and relative humidity. For statistical analyses, we used mean weekly values. We found that 6% of the mean daily concentrations of PM_2.5_ were missing. Weeks with missing data for more than five days were excluded from the analysis. 

### 2.3. Statistical Analysis

The patient and meteorological data collected from January 2013 to December 2019 were sorted, categorized according to weeks, and preliminarily analyzed with a simple correlation analysis (Spearman’s r). Then, the association between meteorological factors, air pollution and the incidence of ILI was further studied with distributed lag nonlinear models (DLNMs) developed by Gasparrini [23]. The DLNM framework enables the modeling of the exposure–response relationship together with the lag–response relationship; thus, it enables the modeling of delayed effects after a specific exposure. In this paper, we used the DLNM to investigate the relationship between the incidence of ILI, meteorological factors and air pollution expressed as the levels of PM_2.5_ and PM_10_. The delayed effects of the abovementioned environmental factors were also studied.

The modeling of the relationship between the incidence of ILI and the environmental factors was performed for the following meteorological factors: air temperature, relative humidity, absolute humidity, sunshine duration, precipitation, precipitation duration, and wind speed and for air pollution expressed as the levels of PM_2.5_ and PM_10_. Poisson regression models with a quasi-Poisson function were used for the analysis to address the issue of the overdispersion of the response data (counts of ILI cases). The number of ILI cases was included in the model as the dependent variable, while the considered environmental factors were included as independent variables. The following formula represents the model structure:log[E(*Y_t_*)] = *α* + *ns*(*X_i_*,*df*) + *ns*(*PM*,*df*,*lag*,*df*) + *ns*(*Time*,*df*)
where E(*Y_t_*) represents the weekly number of ILI cases in week *t*, *α* is the intercept, *X_i_* represents the environmental factors (temperature, etc.), ns is the natural cubic spline, PM is the concentration of particulate matter, df represents the degree of freedom, and Time represents the long-term trend.

Initially, the analysis was performed for each environmental factor separately, and then the models adjusted for air pollutant levels were fitted to the data. Because of the relatively strong correlations between the meteorological factors, one meteorological factor, one air pollutant and the long-term trend were included in each model. The Akaike information criterion for quasi-Poisson function (qAIC) was used to select the degrees of freedom for environmental factors (1–10 df) and for the maximum number of lag weeks included in the model [24]. According to the results of the analysis (lowest value of qAIC) and previous studies [25,26,27], we decided to include the delayed effects for PM_2.5_ and for wind speed. The effects of the meteorological factors, temperature, wind speed and precipitation were modeled using 1 df, sunshine duration 4 df, humidity 3 df and precipitation duration 2 df. For wind speed, the maximum lag of 2 weeks was included (a range of 1–8 maximum weeks of lag). The effects of air pollution caused by PM_2.5_ and PM_10_ were modeled by ns of 1 df, and for PM_2.5_, the maximum lag of 5 weeks was also included. The maximum lag was chosen based on the qAIC and the stability of the shape of the risk of the ILI-PM_2.5_ relationship. The long-term trend was modeled using ns with 1df/year.

The reference levels were defined as the median values of each of the analyzed variables, to calculate relative risks (RRs). The sensitivity analysis was performed by calculating the qAIC and changing df for environmental variables.

The statistical analysis was performed using R software, version 3.6.2, with the packages dlnm, mgcv and bbmle (The R Foundation for Statistical Computing, Vienna, Austria).

## 3. Results

### 3.1. Influenza-Like Illness Activity in 2013–2019

During the analyzed period, 345,987 cases were reported, of which 155,249 (45%) were children under the age of 15, and 31,179 (9%) were individuals over 65 years of age (Appendix A). The incidence rate based on the number of reported cases was 11.4 (95% CI, 11.3–11.5) episodes per 100 person-years. Data on the sex of the patients were unavailable. The median number of cases in a week was 700 (ranging from 0 cases to 6309 cases). The number of ILI cases peaked once in each season, in January and February, when the mean temperatures approached 0 °C (Appendix A). 

### 3.2. Air Pollution and Meteorological Data 

Concentrations of PM_2.5_ and PM_10_ increased in the fall and in the winter (Appendix A). The daily mean concentrations of PM_2.5_ exceeded WHO standards on 767 (30%) days and that of PM_10_ exceeded the standards on 129 (5%) days. The annual mean concentrations of PM_2.5_ exceeded the standards every year, and those of PM_10_ were within the recommended levels in 2019 only (Appendix A). From 2013 to 2019, the weekly mean concentration of PM_2.5_ was 22.41 µg/m^3^, ranging from 9.04 µg/m^3^ to 49.77 µg/m^3^, and that of PM_10_ was 22.34 µg/m^3^, ranging from 6.56 µg/m^3^ to 55.99 µg/m^3^ (Table 1). The weekly mean temperature was 8.28 °C, ranging from –12.90 °C to 23.11 °C. The average weekly relative humidity (RH) was 79.3% (ranging from 45.46% to 97.30%), absolute humidity (AH) was 7.3 g/m^3^ (ranging from 1.34 to 15.91 g/m^3^), wind speed was 2.40 m/s (ranging from 1.05 m/s to 4.66 m/s), precipitation was 1.79 mm (ranging from 0.0 mm to 13.29 mm), precipitation duration was 3.09 h (ranging from 0.0 h to 13.85 h), and sunshine duration was 4.99 h (ranging from 0.0 h to 14.59 h).

### 3.3. Correlations between Influenza-Like Illness Cases and Meteorological Data

The analysis of Spearman correlations showed that the incidence of ILI was positively correlated with weekly mean values of relative humidity, wind speed, precipitation duration, concentration of PM_2.5_ and concentration of PM_10_. In contrast, weekly mean values for temperature, precipitation, and sunshine durations were negatively correlated with the incidence of ILI (Appendix A). When correlations were analyzed in the peak of the flu season only (December–February), ILI cases correlated with PM_2.5_ (R = 0.31, *p* = 0.007), PM_10_ (R = 0.27; *p* = 0.02), mean temperature (R = −0.25; *p* = 0.02), and absolute humidity (R = −0.30; *p* = 0.005). No correlation with relative humidity, wind speed, sunshine duration or precipitation was observed in that time frame.

### 3.4. Non-Linear Univariable Models

For each of the variables described above, a DLNM model illustrating the relationship between the number of ILI cases and the values of the considered variable was fitted. The delayed effect was included for PM_2.5_. The results of the model estimation are presented as relative risks (RRs) and are shown in Appendix A. For each considered environmental variable, the association between the RR of ILI and the values of weekly means of the covariate are shown. As shown in Appendix A, the weekly mean temperature and weekly mean concentrations of PM_2.5_ were the most significant risk factors exponentially associated with the incidence of ILI, as evidenced by the highest relative risks (RRs). The values of the coefficient of determination (R^2^) were 0.523 and 0.565, respectively, which indicates that over 50% of variability in the dependent variable is explained by the models. The RR of the cold effect at the 5th percentile (−5.51 °C) was 2.81 (95% CI, 2.51–3.13) and increased to 4.89 (95% CI, 4.13–5.80) at the minimum value (−12.9 °C) compared to the median (7.99 °C). Compared to the median, the RR of the high air pollution cumulative effect at the 95th percentile of the PM_2.5_ concentration was 13.18 (95% CI, 9.02–19.28), increasing to 19.92 (95% CI, 12.82–30.95) at the maximum value. The RRs of concentrations of PM_10_ were 3.34 (95% CI, 2.66–4.19) at the 95th percentile and 4.08 (95% CI, 3.13–5.32) at the maximum value compared to the median (R^2^ = 0.343). The relative humidity displayed a weak U-shaped relationship with the number of cases. The RRs of the dry and moist effects at the 5th and 95th percentiles, respectively, were insignificant because 95% CIs included a null risk of one. The RR of the relative humidity effect peaked at 90% and reached 1.47 (95% CI, 1.23–1.75; R^2^, 0.256). 

The dry and moist effects were analyzed again using the absolute humidity calculated from relative humidity and temperature. The RR at the 5th percentile of absolute humidity (3 g/m^3^) was 2.61 (95% CI, 2.36–2.88), whereas at the 95th percentile (13 g/m^3^), the RR was 0.24 (95% CI, 0.20–0.28; R2, 0.573). The RR of the high wind speed cumulative effect was 3.20 (95% CI, 2.36–4.34; R2, 0.219) at the 95th percentile (4 m/s). The RR of the effect of low sunshine hours (0 h) was 1.83 (95% CI, 1.36–2.44) and that of the low precipitation (0 mm) effect was 1.37 (95% CI, 1.19–1.57). In contrast, the effect of the precipitation duration was positive and peaked at 9 h (RR, 1.68; 95% CI, 1.34–2.11; R^2^, 0.274); however, that effect was insignificant at the 95th percentile (12 h).

### 3.5. Non-Linear Multivariable Models

Subsequently, the above models were adjusted for PM_2.5_ (Figure 2). The adjusted RRs of the cold effect were 2.29 (95% CI, 2.03–2.58) at the 5th percentile and 3.57 (95% CI, 2.97–4.3) at the minimum value (R^2^ = 0.703). The effect of adjusted PM_10_ was 1.67 (95% CI, 1.19–2.34) at the 95th percentile and 1.82 (95% CI, 1.22–2.69) at the maximum value (R^2^ = 0.548). The highest RR values of relative humidity effects after including PM_2.5_ concentrations decreased to 1.15 (95% CI, 1.05–1.25) at a relative humidity of 86%. When absolute humidity was analyzed, the adjusted RR of the dry effect at the 5th percentile was 2.14 (95% CI, 1.93–2.38). The adjusted RR of the moisture effect at the 95th percentile of absolute humidity was 0.32 (95% CI, 0.27–0.37; R^2^, 0.723). The adjusted RR of the high wind speed cumulative effect was 2.82 (95% CI, 2.20–3.63; R^2^, 0.603). The adjusted RR of the effect of low sunshine hours (0 h) was 1.35 (95% CI, 1.07–1.71; R^2^, 0.616) and that of low precipitation (0 mm) was 1.15 (95% CI, 1.02–1.29; R^2^, 0.576). The adjusted effect of the precipitation duration peaked at 11 h (RR, 1.48; 95% CI, 1.09–2.01; R^2^, 0.585) but was insignificant at the 95th percentile (12 h). 

Considering that temperature is the main driver of ILI of all meteorological factors, the air pollution effect was adjusted for mean temperatures (Figure 2). The adjusted cumulative RRs of the PM_2.5_ concentrations were 3.85 (95% CI, 2.72–5.45) at the 95th percentile and 4.77 (95% CI, 3.19–7.15) at the maximum value. The adjusted RRs of the PM_10_ concentrations were 1.42 (1.15–1.75) and 1.50 (1.17–1.92) at the 95th percentile and the maximum value, respectively (R^2^ = 0.534). Based on the considered DLNM models, the model including PM_2.5_, temperature and long-term data shows the most significant association with the dynamics of ILI prevalence (R^2^ = 0.703). Figure 3 shows the observed and predicted numbers of ILI cases in the considered time period.

Additionally, we considered that PM_10_ reflects the concentrations of both fine (PM_2.5_) and coarse particles (PM_10-2.5_), and therefore, the RRs of the PM_10_ pollution effect were analyzed in a model including the concentration of PM_2.5_ as a covariate. After including PM_2.5_ in the model, the RRs of the PM_10_ effect were 1.67 (95% CI, 1.19–2.34) at the 95th percentile and 1.82 (95% CI, 1.22–2.69) at the maximum value. Furthermore, we assessed the RRs of the cold effect adjusted for the concentrations of both PM_2.5_ and PM_10_. The adjusted RRs of the cold effect were 2.42 (95% CI, 2.01–2.57) at the 5th percentile and 3.54 (95% CI, 2.92–4.28) at the minimum value.

The estimated effects of the PM_2.5_ concentrations were plotted against lag weeks in a model including weekly mean temperatures and a long-term trend as covariates to identify the cumulative effects of PM_2.5_ on ILI cases (Figure 4a,b). The effect of PM_2.5_ decreased over time but remained significant for lag weeks 0 through 4. The RRs (95% CIs) at the 95th percentile were 1.45 (1.31–1.61), 1.37 (1.26–1.48), 1.29 (1.21–1.36), 1.21 (1.15–1.27), 1.14 (1.08–1.21), and 1.08 (0.99–1.16) at lag weeks 0 through 5, respectively. We estimated that a 10 µg/m^3^ increase in the mean weekly PM_2.5_ concentration (an increase from 20 to 30 µg/m^3^) caused a 16% increase in the risk of ILI (RR, 1.16; 95% CI; 1.11–1.21) in the same week (lag 0). That risk decreased over time (Figure 4c). The estimates for the five following weeks (lag 1–5) were as follows: 1.13 (95% CI, 1.10–1.17), 1.11 (95% CI, 1.08–1.13), 1.08 (95% CI, 1.06–1.10), 1.05 (95% CI, 1.03–1.08), and 1.03 (95% CI, 1.00–1.06).

### 3.6. The Effect of Wind Speed

As indicated by the qAIC, we also assessed the effect of wind speed over lag weeks in a model including PM_2.5_ and a long-term trend as covariates (Appendix A). The effect of a high wind speed decreased over time but was significant for lag weeks 0 through 2. The RR (95% CIs) at the 95th percentile was 1.56 (1.32–1.85), 1.41 (1.3–1.54), and 1.28 (1.09–1.5) for lag weeks 0 through 2, respectively.

## 4. Discussion

Air pollution may increase the incidence of a wide range of diseases, including heart disease, stroke, and lung cancer [28,29,30,31]. Numerous studies have confirmed that exposure to ambient air pollutants is strongly associated with the local transmission of respiratory infections [4,5,6,26,32,33,34]. This study provides additional evidence for the association between air pollution and ILI. We have shown that the cumulative effect of an increase in the PM_2.5_ concentration is exponentially associated with the increase in ILI risk in Bialystok, Poland, after adjusting for air temperature and a long-term trend. The distributed lag nonlinear analysis including PM_2.5_, temperature and a long-term trend as covariates showed that the model was able to explain as much as 70% of the variability in the number of observed ILI cases in Bialystok, Poland in 2013–2019. The concentration of PM_2.5_ in Bialystok during the study, frequently exceeded the levels recommended by WHO. However, in our study, the mean PM_2.5_ concentrations were several times lower than those reported in previous studies linking air pollution to ILI [4,26]. Nevertheless, the effect of ambient air pollution on the ILI incidence was evident, indicating that the relationship is not limited to highly polluted regions. Moreover, the high pollution effect extended beyond the incubation periods of influenza and other respiratory tract infections, which are usually shorter than two weeks [35]. Our results show that high concentrations of PM_2.5_ increase the risk of ILI during the same week (lag 0) and for up to 4 subsequent weeks (lag weeks 1–4). Therefore, the underlying mechanisms linking air pollution and the incidence of respiratory infections are not limited to acute effects only.

The leading epidemiological concept of respiratory infections is that infections are spread by direct transmission from person to person, such as touching an infected person or touching the fomites that the infected person has contaminated. Respiratory droplets can also be deposited directly on a person in close proximity to the infected person. However, after droplets are expired, the liquid content starts to evaporate. Some droplets become so small that they are free to travel in the air and carry the pathogens contained inside over tens of meters from where they originated. Previous studies have suggested that airborne transmission should be considered possible in infections caused by influenza viruses [36,37,38,39]. The same might be hypothesized for ILI, even though pathogens other than influenza virus are frequently detected in ILI patients [40]. Multiple studies have suggested that airborne transmission participates in the spread of different respiratory pathogens responsible for ILI like, parainfluenza viruses [41], RSV [42,43], rhinoviruses [44,45], adenoviruses [46,47], human coronaviruses, including SARS-CoV-2 [48,49,50], and other pathogens [51].

Particulate matter may serve as a vector for the transport of pathogenic microorganisms [33,52]. The airborne transmission of pathogens further depends on their survival in aerosolized droplets, which in turn is subject to variations in temperature, humidity, and solar radiation [53]. For example, influenza virus transmission is most efficient under cold, dry conditions [54]. Both correlation and DLNM analyses revealed that temperature, absolute humidity, and wind speed were the meteorological factors that best predicted ILI activity in Bialystok, Poland. Other factors, including precipitation, sunshine duration and relative humidity, showed lower or no correlations with the incidence of ILI. Similar observations were reported in an analysis conducted in other Northern European countries [55]. We found slight inconsistencies when the Spearman analysis was done separately for the peak of the flu season only. This might be explained by the inadequacy of linear analysis in describing the complex relations between weather, air pollution, and respiratory infections. For instance, relative humidity was positively correlated with ILI in a 12-month time frame, but no correlation was found for the 3-month period from December to February. In the nonlinear analysis, an increasing trend towards low and high RH values with a decreased risk was observed at intermediate RH. Thus, our results show that both high and low values of RH facilitate the spread of ILI. The results of experimental studies are consistent with these observations. In laboratory settings, influenza virus was maximally stable at low RH, minimally stable at mid-range RH, and moderately stable at high RH [56,57]. However, humidity is also reported as absolute humidity, defined as the absolute amount of water in the air. We found that absolute humidity was a better predictor of ILI seasonality than relative humidity. Decreasing AH exponentially increased the risk of ILI, even if particulate matter was included in the model. Similar observations were documented in previous studies [58,59]. A positive association between ILI cases and wind speed was also noted. The effect remained significant for up to three weeks after exposure (lag weeks 0–2). Wind was shown to promote the spread of various viruses over short and large distances [60,61]. We hypothesize that wind facilitates the spread of ILI by transporting aerosolized pathogens in the area. The calculated lag effect of two weeks possibly reflects the maximum incubation period of respiratory pathogens causing ILI. Although the wind-borne route alone is insufficient to explain the seasonality of ILI, we show that it contributes substantially to the increase in the risk of infection.

Meteorological variables affect the concentration of particulate matter in the air; however, the interplay between these two factors is complicated. We found negative correlations between PM temperature, absolute humidity, and precipitation. The negative correlation between PM and temperature might be explained by emissions due to residential heating in the cold season [20]. Studies linking humidity and PM have shown conflicting results. In the analysis from the state environmental monitoring program in Poland [20], the authors described significant negative correlations between PM_10_ and RH observed in the warm season from April to September. However, a study from the USA reported a positive correlation between RH and PM_2.5_ concentrations in several regions [62]. These contradictory results may be explained by the relative abundance of each component of the PM. Sulfates and nitrates were shown to be positively related to RH, reflecting in-cloud sulfate formation and the RH-dependent ammonium nitrate formation. The correlation between RH and carbons is negative, which possibly reflects sources from fires and combustion [62]. Moreover, increased RH may be accompanied by precipitation. Rainfall has a washing effect on the levels of PM_2.5_ in the air [63]. In the absence of precipitation, it has been suggested that PM might act as condensation nuclei for water and virus particles [33,52,64].

Notably, pathogens causing ILI differ in structure; thus, meteorological factors probably exert distinct effects on their transmission, making the data difficult to interpret. However, fluctuations in ILI activity in the community are affected not only by the transmission and survival of pathogens in the environment but also by host susceptibility. Exposure to air pollutants impacts the host response to infections. The underlying possible mechanisms related to this process are complex [65]. Ambient air pollution likely disrupts host defenses, including both innate and cell-mediated immune responses against infections. Exposure to PM containing environmentally persistent free radicals (EPFRs), which are found in most combustion-derived PM, was shown to cause pulmonary oxidative stress, which leads to local immunosuppression and the exacerbation of influenza in mice [66,67]. Another study reported enhanced viral attachment and entry due to exposure to diesel exhaust [68]. Particulate matter can also activate inflammatory signaling cascades, change macrophage morphology [69,70] and precipitate tissue remodeling in the lung [71]. Exposure to PM_2.5_ alters the expression of multiple genes in lung tissue [72]. Importantly, the angiotensin-converting enzyme 2 (ACE2) receptor present on the cells was overexpressed following exposure to PM_2.5_. Thus, PM might plausibly increase the probability of SARS-CoV-2 infection, as ACE2 is the key receptor mediating virus entry [73].

The study is subject to some limitations. First, we only used two air pollution measurement sources from Bialystok City, limiting the spatial representativeness of the present analysis. People living outside Bialystok City might have been exposed to slightly different air pollution levels. A vast majority of residents of this area live in towns with similar levels of urban development; therefore, we believe that pollution levels in Bialystok City can serve as a close proxy for the County. Second, our data might potentially suffer from under-reporting of influenza cases. In our opinion, under-reporting tends to occur in consistent patterns throughout the years; hence, the seasonal variation in ILI cases still reflects the true burden of ILI in the region. Third, we did not stratify the relative risks by age group. As previously shown, the effects of air pollution depend on age [26]. Also, it has been reported that transmission of respiratory pathogens is dependent on other factors like population density [74], household size, or income [75], which were not investigated in this paper. Thus, our results cannot be generalized to the entire population, and the calculated relative risks should be interpreted with caution.

## 5. Conclusions

In conclusion, our study describes a clear association between PM_2.5_ pollution and the incidence of ILI in northeastern Poland, despite the relatively low concentrations of particulate matter recorded in the study period. Pollution exerted the greatest effect during the same week it was recorded, but the risk of ILI was increased in the four subsequent weeks. Because SARS-CoV-2 cases are still increasing, further studies elucidating the mechanisms linking air pollution and susceptibility to acute respiratory infections are urgently needed. 

Restrictions imposed to prevent the spread of COVID-19 reduced the concentration of PM_2.5_ in major Polish cities, however, it still remained above the levels recommended by WHO [76]. Globally, the effect of reduced mobility on PM_2.5_ was smaller than expected and probably short-term, because of the relatively small contribution of road traffic to primary PM_2.5_ and the large contribution from secondary precursors of PM [77]. Regardless of the limitations, the above studies clearly show that even short-term changes in anthropogenic activity can improve air quality, which in turn, might affect the susceptibility to ILI. Altogether, our study indicates that all measures implemented to decrease PM_2.5_ concentrations would be beneficial to reduce the transmission of COVID-19 and other respiratory infections.

## Figures and Tables

**Figure 1 viruses-13-00556-f001:**
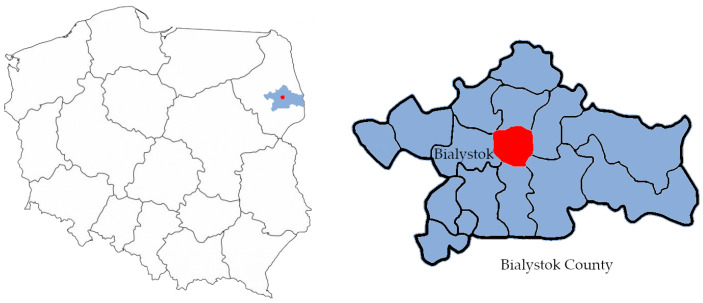
The geographical location of Bialystok and Bialystok County in northeastern Poland.

**Figure 2 viruses-13-00556-f002:**
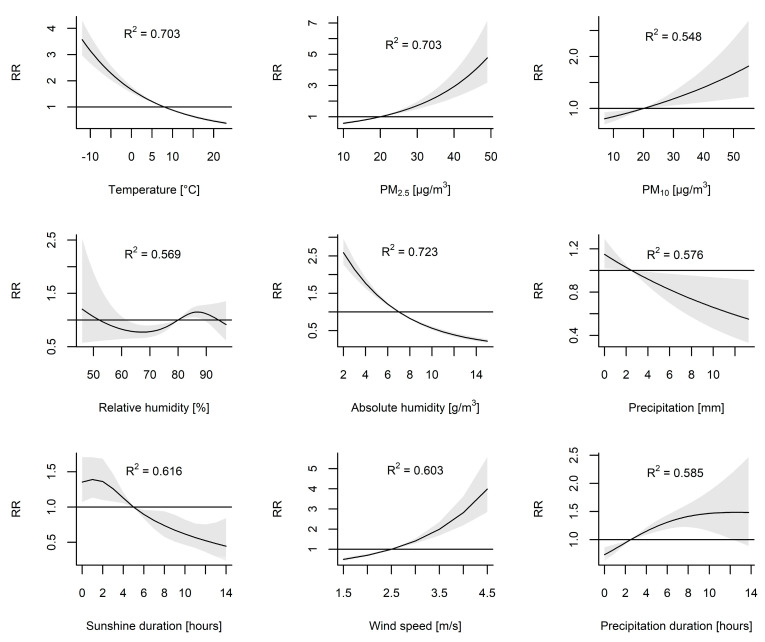
The association between environmental factors and the relative risk (RR) with 95% confidence interval (CI) of influenza-like illness. The models for climatic variables were adjusted for mean concentrations of PM_2.5_ and the long-term trend. The models for PM_2.5_ and for PM_10_ were adjusted for the mean temperature and the long-term trend. The reference levels were defined as the median values of each of the analyzed variables to calculate RRs.

**Figure 3 viruses-13-00556-f003:**
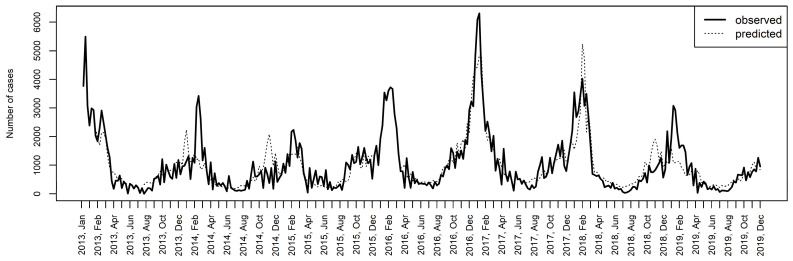
The observed and predicted (according to the model including temperature, PM_2.5_ and the long-term trend as covariates) number of influenza-like illness cases. R^2^ = 0.703.

**Figure 4 viruses-13-00556-f004:**
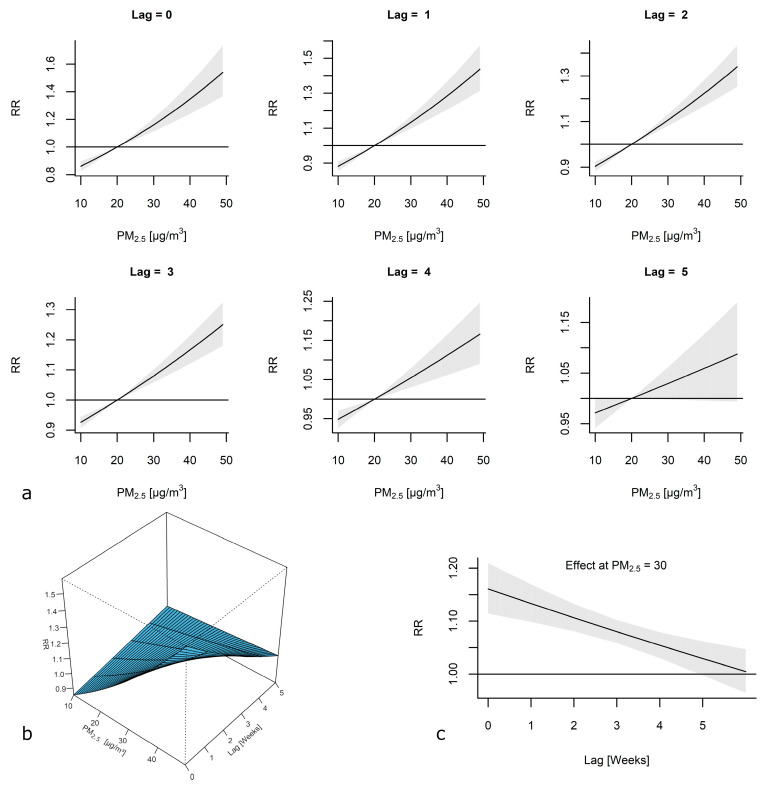
(**a**) The association between the concentration of PM_2.5_ and the relative risk (RR) of the incidence of influenza-like illness (ILI) is depicted as the estimated exposure–response curves for lags of 0 to 5 weeks with 95% confidence intervals (CIs). (**b**) The association between the concentration of PM_2.5_ and the RR of the incidence of ILI is depicted as the estimated exposure-lag-response surface. The model was adjusted for the mean air temperature and a long-term trend. (**c**) The estimated lag–response curve for PM_2.5_ = 30 µg/m^3^ with 95% CI.

**Table 1 viruses-13-00556-t001:** Weekly meteorological variables, air pollutants and the incidence of influenza-like illness in Bialystok, January 2013–December 2019.

	Mean	SD	Min	P5	P25	Median	P75	P95	Max
Number of cases of ILI	1029.72	1034.50	0.00	108.0	313.75	700.00	1280.50	3227.0	6309.00
Temperature (°C)	8.28	8.03	−12.90	−5.51	2.02	7.99	15.83	21.00	23.11
Relative humidity (%)	79.30	9.81	45.46	57.50	72.28	80.51	87.52	96.40	97.30
Absolute humidity (g/m^3^)	7.30	3.18	1.34	2.97	4.65	6.92	10.00	12.79	15.91
Wind speed (m/s)	2.40	0.57	1.05	1.00	2.01	2.33	2.71	4.10	4.66
Precipitation (mm)	1.79	2.09	0.00	0.00	0.29	1.06	2.46	9.82	13.29
Precipitation duration (hours)	3.09	2.75	0.00	0.00	1.00	2.35	4.46	13.40	13.85
Sunshine duration (hours)	4.99	3.75	0.00	0.20	1.42	4.88	7.93	13.16	14.59
PM_2.5_ (µg/m^3^)	22.34	7.96	6.56	7.50	15.26	19.60	26.86	44.70	49.77
PM_10_ (µg/m^3^)	22.41	9.93	9.04	8.60	16.83	20.39	26.90	50.42	55.99

Abbreviations: ILI, influenza-like illness; PM_2.5_, particulate matter with an aerodynamic diameter of 2.5 μm or less; PM_10_, particulate matter with an aerodynamic diameter of 10 μm or less.

## Data Availability

The data presented in this study are available on request from the corresponding author.

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
