# Peer review of "Cumulative Effects of Particulate Matter Pollution and Meteorological Variables on the Risk of Influenza-Like Illness"

_viruses, 2021, doi:10.3390/v13040556_

Round 1
Reviewer 1 Report
The authors present a well-written manuscript detailing the correlation between particulate matter (PM) pollution and the risk of influenza-like illness (ILI) in Poland. The authors seek to use ILI as a proxy to estimate the spread of SARS-CoV-2 and offer the hypothesis that exposure to PM2.5 may increase risk for SARS-CoV-2 infection since it does for ILI. Additionally, they note that PM2.5 exposure alters angiotensin-converting enzyme 2 receptor gene expression in the lungs which is a key receptor for SARS-CoV-2 entry, thus the authors suspect that air pollution may have a role in contributing to the spread of the virus.
The data presented in this manuscript align with many previously reported correlations between ILI and air pollution from countries across the world and in Poland too: https://pubmed.ncbi.nlm.nih.gov/33454853/. Additionally, it has also been reported that air pollution has decreased in many countries across the world due to the shutdowns/quarantines taken to reduce the spread of infection. These findings have also been reported in Poland as well; both PM2.5 and PM10 have decreased significantly during the period of 2018-2019 due to the shutdowns in place to decrease the spread of the virus (https://www.ncbi.nlm.nih.gov/pmc/articles/PMC7657033/). In their conclusions, the authors state that SARS-CoV-2 cases are still increasing. Together, these reports seem contradictory to their current hypothesis that links air pollution to increased susceptibility to SARS-CoV-2 infection.
To strengthen their hypothesis, the authors should fully address these previously reports of decreased air pollution during lockdown and how it relates to their current hypothesis.
Author Response
We would like to thank the reviewer for all the comments. The manuscript has been improved according to the reviewer's suggestions.
The authors present a well-written manuscript detailing the correlation between particulate matter (PM) pollution and the risk of influenza-like illness (ILI) in Poland. The authors seek to use ILI as a proxy to estimate the spread of SARS-CoV-2 and offer the hypothesis that exposure to PM2.5 may increase risk for SARS-CoV-2 infection since it does for ILI. Additionally, they note that PM2.5 exposure alters angiotensin-converting enzyme 2 receptor gene expression in the lungs which is a key receptor for SARS-CoV-2 entry, thus the authors suspect that air pollution may have a role in contributing to the spread of the virus.
We would like to thank the reviewer for this kind review.
The data presented in this manuscript align with many previously reported correlations between ILI and air pollution from countries across the world and in Poland too: https://pubmed.ncbi.nlm.nih.gov/33454853/. Additionally, it has also been reported that air pollution has decreased in many countries across the world due to the shutdowns/quarantines taken to reduce the spread of infection. These findings have also been reported in Poland as well; both PM2.5 and PM10 have decreased significantly during the period of 2018-2019 due to the shutdowns in place to decrease the spread of the virus (https://www.ncbi.nlm.nih.gov/pmc/articles/PMC7657033/). In their conclusions, the authors state that SARS-CoV-2 cases are still increasing. Together, these reports seem contradictory to their current hypothesis that links air pollution to increased susceptibility to SARS-CoV-2 infection.
To strengthen their hypothesis, the authors should fully address these previously reports of decreased air pollution during lockdown and how it relates to their current hypothesis.
Thank you for this comment. The two mentioned papers are important and have been cited in the manuscript. We added a paragraph in lines 429-435.
Our main hypothesis was if particulate matter concentrations and weather conditions affect the risk of all-cause influenza-like illness, which was shown to be the case. The SARS-CoV-2 is just one of many causes of ILI. The COVID-19 cases are still on the rise, which in our opinion is not contradictory to our calculations given the complex relationship between weather, air pollution, the virus, people mobility, and individual susceptibility to infections. We describe just a small part of this relationship and show that PM2.5 increases the risk of ILI and that the effect is long-term. The same might be expected for SARS-CoV-2. However, It cannot be said based on our calculations, that a decrease in PM2.5 after the lockdown should cause a decrease in COVID-19 cases. It might do so, but there are so many other factors influencing the transmission of the virus that the sole effect of brief reduction in air pollution on COVID-19 would be difficult to calculate.
Reviewer 2 Report
the paper is methodologically correct, therefore, I proposed minor revisions. I would appreciate considering the improvement of "limits" section as suggested in the comments

Author Response
We would like to thank the reviewer for all the comments. The manuscript has been corrected according to the suggestions.
General comment
Knowledge on the subject is still unclear and, in some cases, inconsistent. Therefore, I appreciated the goal of investigating the effect of exposure for additional temporal dimension. As for this, the study, offers a deepening of current knowledge. The paper is well written and the use of statistical tools is
appropriate. The use of a large cohort, to follow the trend of the incidence, is an advantage to stabilize the results with respect to seasonality. Some minor revisions are proposed to complement current aspects of discussion on the topic.
Thank you for this kind review.
Minor revisions
Line 52: “Based on accumulating evidence, PM increases the incidence of influenza and influenza-like illness [4–6].”
Please, could you provide arguments about the different behaviour in transmission of PM10 and PM2,5? Other studies observed that PM2.5 only was consistently associated with increased illness-like viruses incidence and that, increased PM10 concentration was associated with decreased illness-like viruses
incidence, the phenomenon seemingly be associated to a diameter-related effect. (as discussed in: Jiang Y, Wu XJ, Guan YJ. Effect of ambient air pollutants and meteorological variables on COVID-19 incidence. Infect Control Hosp Epidemiol. 2020 Sep;41(9):1011-1015. doi: 10.1017/ice.2020.222. Epub
2020 May 11. PMID: 32389157; PMCID: PMC7298083.)
Thank you for this suggestion. We have added the suggested citation in lines 53-54. However, the detailed discussion and analysis of different effects of PM2.5 and PM10 was beyond the scope of this work. Influenza-like illness is caused by different pathogens of different sites of entry and the sole capability of PM2.5 being a potential vehicle for viruses transporting them deep into the lungs in our opinion is just one of many explanations. Influenza A viruses (which are responsible for the majority of ILI cases), for instance, bind to receptors in the upper respiratory tract. Therefore the fact that PM2.5 might reach alveoli does not fully explain the difference between the risk imposed by PM10 and PM2.5 on ILI.
Line 201: “No correlation with relative humidity, wind speed, sunshine duration or precipitation was observed in that time frame”.
Consider including the explanation for inconsistency with findings in different time.
The linear analysis was just a first step to get some insights into possible relations between the variables. The observed inconsistency was probably caused by the inadequacy of linear analysis in describing the complex relations between weather, air pollution, and respiratory infections. That explanation was added in lines 347-352.
Lines 412-413: “Pollution exerted the greatest effect during the same week, but the risk of ILI was increased in the four subsequent weeks. Because SARS-CoV-2 cases are still increasing, further studies elucidating….”
The study deals with a complex phenomenon and investigates only a part of the factors that can play a role in transmission and diffusion. Since it is a viral infection, subjected to transmission mechanisms through contagion other than those that characterize the spread of atmospheric pollution, other factors
must be mentioned. The analysis model explains an effect of PM2,5 and relative humidity of a certain importance, however, there is no reference to other potential factors that, if considered, can make the model more consistent with reality, like the following:
ï‚· it has been documented that transmission outbreaks arise in confined areas where the
population density is higher regardless of the more or less high levels of pollution;
ï‚· it has been speculated that in some areas, where there is a high mobility of workers, a greater
transmission is observed due to the use of different modes of transport that offer greater
conditions for the transmission of infections;
ï‚· not negligible factors, in the transmission of influenza diseases, are the housing condition and
the level of social deprivation. The worsening of these two factors was associated with a
significant increase in indoor transmission
ï‚· general health status of the population is relevant basic condition, even though, the authors
recognize the general importance of individual susceptibility
Thank you for this remark. We think that the heterogeneity in the general population as well as in pathogens causing ILI makes it complicated to detect associations between potential risk factors and respiratory infections. It it truly important to analyze the phenomenon in a broader context and we should definitely plan future studies addressing this issues. We have added a sentence in the limitations section.